# The Role of Thermal Water in Chronic Skin Diseases Management: A Review of the Literature

**DOI:** 10.3390/jcm9093047

**Published:** 2020-09-22

**Authors:** Sara Cacciapuoti, Maria A. Luciano, Matteo Megna, Maria C. Annunziata, Maddalena Napolitano, Cataldo Patruno, Emanuele Scala, Roberta Colicchio, Chiara Pagliuca, Paola Salvatore, Gabriella Fabbrocini

**Affiliations:** 1Department of Clinical Medicine and Surgery, University “Federico II” of Naples, Via Pansini 5, 80131 Naples, Italy; sara.cacciapuoti@libero.it (S.C.); mat24@libero.it (M.M.); marica.annunziata@hotmail.it (M.C.A.); emanuele.scala@outlook.com (E.S.); gafabbro@unina.it (G.F.); 2Department of Public Health, University “Federico II” of Naples, Via Pansini 5, 80133 Naples, Italy; 3Department of Medicine and Health Sciences “Vincenzo Tiberio”, University of Molise, 86100 Campobasso, Italy; maddy.napolitano@gmail.com; 4Department of Health Sciences, University Magna Graecia of Catanzaro, 88100 Catanzaro, Italy; cataldopatruno@libero.it; 5Department of Molecular Medicine and Medical Biotechnologies, University of Naples Federico II, 80131 Naples, Italy; roberta.colicchio@unina.it (R.C.); chiara.pagliuca@unina.it (C.P.); psalvato@unina.it (P.S.)

**Keywords:** thermal water, chronic skin diseases, psoriasis, atopic dermatitis

## Abstract

The benefits of thermal water in different diseases have been known since ancient times. Over the past decades, a re-assessment of the use of mineral water for the treatment of several pathologic conditions has taken place around the world. Today, water therapy is being practiced in many countries that have a variety of mineral springs considerably different in their hydrogeologic origin, temperature, and chemical composition. Thermal water and balneotherapy offer several advantages: this approach needs no chemicals or potentially harmful drugs; there are almost no side effects during and after treatment, and there is a low risk to the patient’s general health and well-being. However, it is difficult to evaluate the efficacy of this therapeutic approach in clinical practice due to the complexity of molecular mechanisms underlying its efficacy. Here we review the current knowledge of the chemical, immunological, and microbiological basis for therapeutic effects of thermal water with a specific focus on chronic inflammatory skin diseases. We also describe recent evidence of the major dermatologic diseases that are frequently treated by balneotherapy with a remarkable rate of success. Moreover, we discuss the potential role of balneotherapy either alone or as a complement to conventional medical treatments.

## 1. Introduction

The role of the balneotherapeutic effects of spring water on skin diseases has been known since ancient times [1]. “Ut mulier suavissima et planissima fiat et sine pilis a capite inferius, in primis eat ad balnea.” (“The woman must go to baths first to have a soft and velvety skin and remove hair from all parts of the body”) was the incipit of “The ornatu mulierum”, probably the first treatise of cosmetic of history of Trotula de Ruggiero (XI century) [2]. In the fourth century B.C. in the first known medical work (Corpus Hyppocraticum), the theme of water was treated from a scientific point of view. Later in history men were discovering the beneficial properties of water, such as its healing and disease-protecting effects. Although it was because of the Greeks that thermalizm was born, it experienced its golden age in the Roman Era and thermal baths were considered as a regular regimen for health: hydrology became a real science and thermal treatments were prescribed with specific indications to follow and underwent medical surveillance [3,4,5,6]. In more recent times, after a decline in popularity, therapeutic properties of water in the forms of balneotherapy and spa therapy emerged, first in Europe and then in the United States [7]. At first, cosmetic properties of thermal waters gained popularity and the cosmetic industry marketed thermal spa waters as cosmeceuticals, expanding the skincare armamentarium. Later, scientific evidence demonstrating the therapeutic properties of thermal waters in several pathologic conditions increased. In 2002, the World Health Organization (WHO) established the Pain Management Protocol for the relief of pain in the document “WHO Traditional Medicine Strategy 2002–2005”, recognizing the efficacy, and quality of Integrative and Complementary Practices (PICS) including balneotherapy, hydrotherapy, and chronotherapy [8]. Thermal medicine is currently considered the medical branch that uses the thermal water properties with curative and the rehabilitative purposes and its efficacy are well documented in different dysfunctions, from rheumatic diseases [9] to cardiovascular ones [10]. Furthermore, different studies have shown that thermal water acts on cutaneous pathologies and skin regeneration [11], also used in cosmetics formulations for their anti-irritant effects [12]. Dermatologic diseases frequently treated by balneotherapy with a high rate of success are psoriasis (PSO) and atopic dermatitis (AD). Other conditions treated by balneotherapy include acne vulgaris, lichen planus, pruritus, rosacea, seborrheic dermatitis, and xerosis. The mechanisms by which broad spectrums of disease are alleviated by thermal water have not been fully elucidated. It is known that thermal medicine is a clinical complementary approach in the treatment of low-grade inflammation and stress-related pathologies. However, the biological mechanisms by which the use of thermal water alleviate symptoms of several pathologies are still not completely understood. Neuroendocrine and immunological responses triggered by balneotherapy are involved in its therapeutic activity, thanks to anti-inflammatory, analgesic, antioxidant, chondroprotective, and anabolic effects together with neuroendocrine-immune regulation. Depending on the thermal waters’ composition, they might have a target action towards chronic inflammatory skin disorders including PSO and AD. In this review, we focus on current knowledge of the chemical, immunological, and microbiological basis for therapeutic effects of thermal water in chronic inflammatory skin diseases. We also describe principal evidence both on the major dermatologic diseases that are frequently treated by balneotherapy with a remarkable rate of success and other skin diseases that can potentially benefit from this therapeutic option. Moreover, we discuss the potential role of balneotherapy either alone or as a complement to other therapies to be considered after, or accompanying, conventional medical treatments [13,14,15].

## 2. Materials and Methods

This study is a narrative review. Pubmed and Scopus databases were reviewed up to May 2020. Search terms included “balneotherapy”, “spa therapy”, “mineral water”, “mineral spa”, “therapeutic spa”, “sulfurous water”, “spring”, “seawater”, and “psoriasis” and “atopic dermatitis” in the abstract and title. The obtained articles were evaluated for scientific evidence related to chronic skin diseases. Duplicate articles, those with no full text available, letters to the editor, and case reports were excluded. In addition, articles about balneophototherapy were excluded to not analyze the further additive effects of sunlight or artificial UV to thermal water. Finally, the findings were analyzed by content analysis and compared further.

## 3. Thermal Water in Dermatology

### 3.1. Chemico-Physical Properties

Water therapy is practiced in many countries with a variety of mineral sources that are different in their chemical and physical properties. In 1933, Marotta and Sica [16] classified mineral waters according to three parameters: temperature, fixed residue, and chemical composition.

Temperature is the first property that mineral waters acquire during their origin. It is determined by several conditions including geothermal gradient, ascent speed, and exothermic reactions. Thermal waters can originate with magma (“juvenile waters”) or they can contact it during their ascent [17]. Magma is between 650 and 1200 °C and the heat gives to the water hydrogen bond a great energy charge, which is released during cooling [18]. In addition, the individual particles in solution acquire movement called “thermal noise”: waters can transmit these vibrations to the surfaces in contact with them [19,20]. According to temperature derived from energy potential, waters are classified in: cold (temperature below 20 °C), hypothermal (between 20 and 30 °C), homeothermal (between 30 and 40 °C), and hyperthermal (between 40 and 50 °C).

During the process of ascent, thermal waters acquire different characteristics. In fact, claimed beneficial effects are supposed to derive from waters’ specific properties: fixed residue and chemical composition. Fixed residue mg/L at 180 °C is the total amount of inorganic solvent. Fixed residue is one of the most important parameters allowing for the official classification of waters with therapeutic properties. Chemical composition is defined by the presence of anions and cations. When an ion is present in quantities greater than 20 meq/L it gives the name to the water. The presence of several predominant ions can define a classification of multi-ion waters. Classification of natural mineral waters based on temperature, fixed residue at 180 °C, and chemical composition have been reported in Table 1 [21,22,23].

Hydrotherapy can have several applications. For example, cold and hypothermal waters cause physiologic reactions such as a decrease in local metabolic function, local edema, nerve conduction, muscle spasm, and an increase in local anesthetic effects [24]. In dermatology, the use of a cold pillow compress has been proposed to reduce hair follicle damage and consequent alopecia caused by chemotherapeutic agents [25]. In the same way, thermal stimulation causes vasodilatation, enhances blood circulation, and decreases blood pressure. Hyperthermia also has important effects on granulocyte mobility and microbial and enzymatic activities. Other beneficial effects of thermal stimulation include increased extensibility of collagen-rich tissues [26,27]. On the other hand, hyperthermal waters are used as short-term thermal stress: human skin can release significant amounts of opioid peptides, modifying the threshold of pain [28]. 

Several therapeutical effects of thermal water are due to a special combination of chemical elements depending on the hydrogeologic origin of each thermal source. In dermatology, thermal waters are chosen for different indications according to salts concentration [13,29]. For example, thermal water used in cosmetics is hypertonic [30]. The high concentration of minerals is easily assimilated by the skin improving its physiological functions. The large presence of minerals is the main feature that makes thermal water effective for the well-being of the skin. The characteristics of thermal waters used in dermatology are reported in Table 2. 

Mineral waters (in particular salty and sulfur waters) are considered particularly useful for therapeutical applications in dermatology due to their keratolytic, regenerative, and antioxidant effects [31,32,33]. Furthermore, bathing in mineral waters at different temperatures can remove microbial peptides that cause many skin diseases [34], reduce inflammation, improve microcirculation, regulate immune processes, and increase the quality of life [35,36,37,38]. As demonstrated in several clinical studies, these beneficial effects of thermal waters are related either to the skin absorption of mineral elements or to skin temperature regulation [39,40,41,42,43,44]. In conclusion, the physical and chemical composition of thermal waters significantly contributes to their therapeutic activity and they must be carefully selected when prescribing a therapy for different dermatologic diseases.

### 3.2. Immunomodulatory Effects of Thermal Waters

Anti-inflammatory and immunomodulatory effects have been attributed to thermal baths and spa therapy [1,23]. The heat stimulates the body to release a range of immunomodulatory mediators such as β-endorphin, enkephalin, and irisin [1,42,43,44]. In particular, β-endorphin and enkephalin influence pain perception and regulate the proliferation of immune cells [42,43], whereas irisin ameliorates the overall metabolic status and cognitive capacities [44]. Moreover, spa therapy has been shown to increase blood flow by dilating capillaries and decrease fibrinogen concentration [40,45]. 

Apart from this, body exposure to thermal waters stimulates the immune and antioxidant systems [46,47]. Cumulative studies have reported that the application of minerals like sulfur, manganese, magnesium, zinc, selenium, strontium, silica, and calcium bicarbonate might have immunomodulatory effects on skin disorders including AD, contact dermatitis (CD), and PSO [15,48,49,50,51,52]. Among the minerals, sulfur can dose-dependently inhibit T-cell proliferation and cytokine production such as interleukin (IL)-2, IL-8, IL-23, IL-17, and interferon (IFN)-γ. It also impairs keratinocyte cell growth and adhesion inhibiting mitogen-activated protein kinase signaling [15,53,54,55,56]. Both sulfur and manganese have bactericidal activity against *Staphylococcus aureus* (*S. aureus*) which is commonly detected in AD, and hidradenitis suppurativa (HS) lesions [48,57,58]. Magnesium and zinc reinforce the skin barrier and immune system, whereas the combination of magnesium and calcium salts accelerate skin recovery. An in vitro study has shown that selenium and strontium inhibit cytokine production, mainly IL-6, by epidermal cells. Further, selenium can suppress inflammatory response by Langerhans cells (LCs) [59]. Concerning silica and calcium bicarbonate, they inhibit mast cell histamine release and decrease cutaneous basophil degranulation potentially preventing the itch–scratch cycle in AD [52]. Additionally, the combination of ions such as magnesium, calcium, chlorine, manganese, sulfur, and strontium reduces the AD-like inflammation in hairless mice via immune-modulation and redox balance [47]. Particularly, a reduction of AD signature cytokines such as IL-1β, tumor necrosis factor (TNF)-α, and Th2 cytokine IL-13 was reported. 

Therefore, the therapeutic effects of thermal waters (salty or spring ones) on the skin might depend on the concentration of the aforementioned minerals. As reported in a review by Khalilzadeh et al. [23], salty thermal sources highest in minerals can reduce: (i) the human leukocyte elastase enzyme (involved in PSO), (ii) the transforming growth factor (TGF)-β (which is increased in psoriatic patients), (iii) the LCs of the skin, (iv) the aging markers, and (v) the skin infections, through the removal of yeasts and bacteria which classically contribute to seborrheic dermatitis (SD). Regarding spring thermal sources, the regulation of immunomodulatory parameters by spa water-supplemented media was observed in human psoriatic keratinocytes (Comano Thermal Water—CTW, Italy) [60], LCs (La Roche Posay-LRP, France) [61], mast cells (Avène Spa Water-ASP, France) [62], and CD4+ T lymphocytes (ASP, France) [63]. Most notably, TNF-α and IL-8 production was reduced in psoriatic keratinocytes by CTW [60], and a partial shift from a Th2 to a Th1 cytokine profile was observed by ASW [63], offering a rationale for the treatment of PSO and AD, respectively. Preliminary studies using cultured fibroblasts suggest enhanced plasmatic membrane fluidity by ASW [64]. Moreover, differentiation of skin keratinocytes as measured by the expression of involucrin, and cytokeratins-1 as well as cytokeratins-10 was induced by ASW [65]. Nevertheless, it is difficult to extrapolate thermal water efficacy in clinical practice due to the complexity of molecular mechanisms underlying skin disorders. It should be kept in mind that the therapeutical effects of thermal waters are due to the combination of chemical, physical, and microbiological characteristics. Therefore, depending on the thermal waters’ composition, they might have a target action towards chronic inflammatory skin disorders including PSO and AD. More evidence is necessary to identify which thermal waters are more appropriate to treat certain skin disorders rather than others.

### 3.3. Microbiological Properties

Since 1998 there has been a growing interest in the characterization of the microbial community of thermal waters [66,67]. On one side, the microbial evaluation of water for human use is subjected to sanitary control, in order to search for coliform bacteria, *Salmonella* spp., and fecal *Streptococci* [68,69] and on the other side, the analysis of the microbial composition of the thermal waters revealed the presence of dynamic microbial communities that function in different microenvironmental conditions [70,71,72,73], made up of peculiar and unusual microbial species with potential therapeutic effect [74,75,76]. Several environmental factors shape microbial communities that inhabit thermal water like temperature, water chemical composition, and pH as well as the distance from the sources [71]. Research on the alkaline–silica hot spring of Yellowstone National Park (USA) highlighted that these microbial communities are controlled primarily by environmental temperature. In particular, the community similarity decreases exponentially with increasing differences in temperature between samples but was only weakly correlated with physical distance from the source and the diversity wanes with increasing temperature but was uncorrelated with other measured environmental variables [70]. Comparable results were obtained on Nakabusa thermal spring water (Japan): terminal restriction fragment length polymorphism (T-RFLP) profiles of bacterial 16S rRNA genes revealed that complexity in the community increases as temperature decreases [71]. Even in hot springs situated in the Western Plain of Romania, the temperature was found to exert a strong control on alpha diversity, with species richness and evenness proportionally dropping with the increase in temperature. Conversely, the beta-diversity was found to be influenced to a greater extent by the physicochemical characteristics of each hot spring water, especially by the dominant ion concentrations (Na+, K+, HCO3-, and PO43), temperature, and electrical conductivity [72]. Therefore, microenvironmental conditions of thermal spring water shape microbial communities that make critical contributions to ecosystem functions through their participation in biogeochemical cycles [70]. Regarding the hottest and hot thermal spring waters, Aquificales represent the thermophilic bacterial lineage most prevalent at high temperatures (over 66 °C) [71,77,78,79]. Wang et al. reported that Aquificae occurred at high-temperature springs, and their distribution was directly proportional to temperature and silica contents [80]. Whereas, at cooler temperatures (<60 °C), moderate thermophilic and mesophilic bacteria such as, *Thermosynechococcus andSynechococcus* (*Cyanobacteria*), and *Chloroflexaceae* became predominant [70,71,80]; the phylum Deinococcus–Thermus, is dominated by the genera *Thermus* with some *Meiothermus.* Species of *Thermus* grow with an optimal growth temperature of 65–75 °C, whereas *Meiothermus* spp. have a lower optimal growth temperature 50–65 °C [71,78,79,80]. Among hot springs, Proteobacteria grow at the lowest temperature (about 60 °C) which may be a feasible temperature for the growth of these bacteria [77,79]. In agreement with this consideration, in warm thermal spring waters (20–35 °C), high detection frequency of bacteria belonging to the Proteobacteria phylum (alpha, beta, and gamma) was observed [73,81,82], followed by members of Actinobacteria [73,81,82], Bacteroidetes [81,82] (Table 3).

More recently, based on the phenotypic, chemotaxonomic, phylogenetic, and genetic characteristics, novel bacterial species from thermal spring waters were identified. In particular, a novel species of *Desulfovibrio* was isolated from the waters of a Tunisian thermal spring, proposed as *Desulfovibrio biadhensis* [74]. Moreover, from a radioactive thermal spring in Budapest, a novel species of the genus *Deinococcus* was isolated, named *Deinococcus fonticola* [83], and from thermal spring water sampled at Xi’an (PR China) a novel species of a new genus in the family Chitinophagaceae, phylum Bacteroidetes, for which the name *Paracnuella aquatica* was proposed [84]. 

Further studies focused their attention also on the metabolic potential of microbiota associated with thermal spring water to identify a promising source of enzymes with biotechnological potential [85,86]. While the chemical concentrations in thermal waters are admittedly associated with their therapeutic effects [87], the inclusion of efficient bioproduct additives produced by photosynthetic organisms and which act against oxidative stress may comprise a significant supplementary value for the increasingly competitive sector of balneotherapy [76], such as a Cyanobacterium *Leptolyngbya sp.* that possesses abundant natural antioxidant products which may have prophylactic and therapeutic effects on many types of illness and toxicity [76]. In addition, an original microorganism, *Aquaphilus dolomiae*, never described in another medium, has very recently been identified in the Avene thermal spring water. *Aquaphilus dolomiae* was found to be responsible for significant pharmacological activities on inflammation, pruritus, and enhancement of innate immunity [88]. The complex diversity and composition of microbial communities on the skin vary by skin region [89] and between individuals [89]. The skin microbiota is composed of around 80% Gram-positive and 20% Gram-negative bacteria. Several diseases such as atopic dermatitis and psoriasis have been found to be associated with changes in the composition of the skin microbiota. Balneotherapy using probiotic La Roche Posay Thermal Spring water has been shown to improve the skin microbiota in a variety of inflammatory skin conditions [90]. More recently, natural mineral water itself revealed antimicrobial properties against the main pathogens responsible for several skin disorders [91].

## 4. Thermal Water and Chronic Inflammatory Skin Diseases

### 4.1. Psoriasis

Thermal balneotherapy (TB) and balneophototherapy (BPT), where thermal mineral baths are combined with natural or artificial ultraviolet (UV) light, have been described as efficacious treatments in PSO [1,39]. Particularly, even if TB and BPT became less popular experiencing a decline among both patients and physicians in the last decades for the advent of rapid and efficacious treatments such as biologics, they have minimal side effects such as skin irritation and itching [1,39]. Moreover, TB does not interfere with metabolic comorbidities and does not cause drug interactions even if acute infection, active tuberculosis, poorly controlled arterial hypertension, and cardiac arrhythmias contraindicate its use as the case of pustular or erythrodermic psoriasis. PSO is a chronic skin inflammatory disease whose natural course may be strongly influenced by weather, humidity, temperature, and climate factors in general [92,93]. Many authors report that TB alone or BPT can give major benefits to PSO patients (mainly plaque PSO) after three to four weeks of treatment [1,39,94,95,96,97,98]. However, only a few controlled clinical trials on their efficacy are available, mainly focusing on BPT [99]. However, the current review will only focus on the role of BT in PSO treatment, not analyzing the further additive effects of sunlight or artificial UV to thermal water. Among BTs, BT at Comano spa in Trentino, Italy, has long been studied and used in the treatment of PSO [60,100,101,102,103,104]. Comano water is an oligometallic thermal water, containing various microelements, among which calcium and magnesium are more represented; it has a temperature of 27 °C in the springs and a pH of 7.5–7.6. Several studies have shown that the exposure of cultured human psoriatic keratinocytes to Comano water in vitro significantly down-modulates the expression and secretion of vascular endothelial growth factor-A (VEGF-A) isoforms, interleukin (IL)-6, IL-8, tumor necrosis factor (TNF)-α, and cytocheratin-16. Hence, BT at Comano spa may be efficacious in PSO by reducing the level of several cytokines and chemokines involved in PSO pathogenesis [60,102,103]. Indeed, Peroni et al. showed that one-week or two-week balneotherapy resulted in a significant reduction in PSO area severity index (PASI) score (11.54 ± 2.76% and 13.5% ± 23.1%) among 77 patients with mild to severe PSO [39]. The authors also reported similar results as regards the patient’s reported PSO severity evaluation and quality of life outcomes. Despite being well tolerated with only mild to moderate cutaneous discomfort (pruritus, burning sensation, and skin dryness) reported effects, during the following three months, many patients experienced a progressive worsening of their skin condition and quality of life with a return to the basal situation. However, a previous randomized, double-blind study had shown a higher efficacy of TB with spring Comano water compared to balneotherapy with tap water [104]. Bathing in the Blue Lagoon, a specific geothermal biotope in Iceland, has been known for many years to be beneficial for human skin in general and for patients with PSO in particular [94]. The chemical composition of the fluid in the lagoon is complex involving silicon monoxide, sodium, calcium, potassium, and chlorine above all with a mean temperature of 37 °C; the mean pH is 7.5, and the salt content is 2.5%. In 1992, 27 PSO patients (body surface area >10%) bathed at Blue Lagoon three times a day for one hour at a time for three weeks [105]. The PASI score significantly decreased from 16.1 to 10.8 after one week and to 8.1 after three weeks, with 19.2% of patients reaching at least an improvement of 75% form their initial PASI score after three weeks. The authors also observed that the area of the lesions did not diminish but scaling erythema and infiltration decreased. Further data on TB for PSO came from Bulgaria, particularly from the Jagoda spa water which is thermal (42.9 °C) and slightly mineralized (556.35 mg/L) with the major components being sulfate, hydrocarbonate sodium, and potassium, and HSiO [98]. In the period from 1984 to 1986, 41 PSO vulgaris patients were sent from the Department of Dermatology, Medical University at Sofia for TB in the Jagoda spa, with 79% being discharged three weeks later with major improvement or clinically healthy [92]. Historically, the beneficial effect of TB in sulfuric rich spas on PSO has been described also in Korean and Polish spas [95,106]. France is another important country for TB. La Roche-Posay is a small town in the center of France that is famous for its dermatologic thermal treatment centers, being the largest thermal center for dermatologic diseases in the world [1]. La Roche-Posay is a selenium-rich water (0.053 mg/L) that contains bicarbonate, calcium, silicate, magnesium, and strontium, and is famous for its anti-inflammatory and antioxidant proprieties [107]. TB consists of an 18-day treatment (three weeks) with the daily application of a high-pressure filiform shower (15 bars for three minutes), performed by a dermatologist [1]. Martin et al. observed that moderate to severe PSO patients that underwent TB at La Roche-Posay experienced a decrease in disease severity together with improved bacterial biodiversity among non-lesional and lesional skin [105]. Indeed, Xanthomonadaceae belonging to the Proteobacteria phylum are significantly increased, and since they are keratolytic, their augmented level has been associated with PSO clinical improvement. Moreover, a study performed in 1995 with 92 patients with moderate PSO found a significant PASI reduction by 47% after three weeks of TB at La Roche-Posay with 8% of patients being completely clear and 48% improving by more than 50% [108]. The efficacy of TB for PSO is further underlined by other randomized controlled trials [42,109]. Particularly, Salies de Béarn saline spa water alone (sodium concentration, 250 g/L; magnesium concentration, 980 mg/L) (one treatment a day, five days a week for a total of 21 days) was able to reduce PASI score by 29%, however, being less efficacious in respect to nb-UVB alone or with TB combination [109]. Furthermore, Levico and Vetriolo (Trentino Region, northern Italy) arsenical-ferruginous water from Italy (pH 1.6, with a high concentration of iron and sulfate and trace arsenic) was shown to produce a statistically significant decrease in mild to moderate PSO lesions in 34 patients with daily 20-min wet packing for 12 consecutive days [42]. A statistically significant difference between spa water-treated lesions and placebo-treated lesions in the same patients was demonstrated for histopathologic (a composite score of hyperkeratosis, parakeratosis, presence or absence of the granular layer, acanthosis, papillomatosis, mitoses, presence of dilated blood vessels, neutrophilic exocytosis, dermal lymphocytic inflammatory infiltrate, and the presence of neutrophils in the infiltrate) and immunohistochemical parameters (proliferation antigen Ki67). It was hypothesized that high concentrations of arsenic in the water could induce keratinocyte apoptosis in psoriatic lesions; thus, TB can modestly improve PSO. TB performed in Copahue Thermal Basin Complex, Argentina, combining daily bathing with volcanic water and application of mud or algae from the sulfurous lagoons, has also shown to be effective in PSO treatment [110]. Particularly, thermal therapy comprised of two daily baths, one with thermal water (algae rich “Green Lagoon” with volcanic water) and the other with mud and/or algae from the sulfurous lagoons with all PSO patients (*n* = 55) experiencing an improvement in scaling and worsening of erythema at day five and marked improvement in erythema, scaling and xerosis at day 10, together with pre- and post-treatment reported histological skin features changes. Thermal bath plus muds have been shown to reduce PSO severity also in Ischia island, Italy, with four weeks of therapy reducing PASI from 10 to 6 among 18 patients [111]. Even if not supported by a single study or trial, many different centers have been recommended for TB skin diseases, PSO above all, involving different countries such as Italy, Spain, Germany, Poland, Bulgaria, and Serbia as well as the USA (Spa resorts of Saratoga Springs (New York), Hot Springs (Arkansas), White Sulfur Springs (West Virginia), and Bedford Springs (Pennsylvania) [112]. In addition, bathing in the Dead Sea (located at the Israel and Jordan border) for PSO treatment has been extensively studied in literature, proving to be an effective and promising strategy [1,97,112]. Even if the Dead Sea also shows thermal qualities, the unique spectrum of UV light (greater in UVA and very low in the UVB portion of the spectrum compared to other locations, permitting a prolonged exposure to the sunlight with minimal phototoxicity) present at this unique and peculiar location (about 400 m below sea level, with the highest concentration of salts and minerals) is thought to be the principal component responsible for clearing the skin. For these reasons, articles evaluating bathing in the Dead Sea have not been examined in this review. In conclusion, TB has been shown to be able to decrease scale, pruritus, and inflammation in PSO patients, appearing to be an interesting adjuvant or alternative therapy especially for patients willing to discontinue momentarily pharmacologic therapy. However, studies reporting positive efficacy of TB on PSO are very heterogenous for disease severity, treatment regimens, and duration, so standardized criteria and algorithms for TB in PSO management are lacking. Hence, even if TB is a natural therapy that has been used around the world for many centuries, it remains an adjunctive or alternative treatment that may be offered only to a selected and limited class of PSO patients, also because its effects are lost in few months.

### 4.2. Atopic Dermatitis

AD is a chronic inflammatory skin disease affecting up to 20% of children and up to 10% of adults [113]. Quality of life is highly impaired due to pruritus, poor quality of sleep, and skin lesions located in visible areas [114]. The pathogenesis of the disease is still elusive, including both genetic (disruption of skin barrier and hyperactivity of Th2 immunity) and environmental factors. The diagnosis is essentially clinic, especially in adults where some atypical phenotypes can be found [113]. The treatment is essentially based on the use of topical emollients and anti-inflammatories, such as corticosteroids, inhibitors of calcineurin; in moderate-to-severe cases systemic treatment is needed (phototherapy, cyclosporine A, or dupilumab) [115]. Balneotherapy (BT) has been widely used as an ancillary therapeutic tool for the treatment of chronic inflammatory skin diseases; indeed, its efficacy has not been studied systematically in AD [115]. In the current reports, BT was investigated as part of a complex climatotherapy or in combination with ultraviolet (UV) therapy [111]. Clinical trials at the Dead Sea have reported the positive effects of BT on AD [97,116,117,118,119,120,121,122]. Four variables contribute to the Dead Sea’s balneological qualities: atmospheric, solar, chemical, and thermal [1]. Favorable atmospheric conditions include low precipitation, almost cloudless skies approximately 330 days a year, moderate to high temperatures, low humidity, very high barometric pressure leading to 10% increased oxygenation of the air compared with sea level, and low pollen count [1]. Due to its low elevation, ultraviolet light from the sun is attenuated at the Dead Sea, contributing to an overall low radiation level and a higher ratio of UVA rays (320–400 nm) to UVB rays (290–320 nm). In addition, the high salt and mineral content of the Dead Sea and high evaporation rate create a continuous haze of aerosols containing magnesium, calcium, potassium, and bromine above the body of water, further filtering the shorter UV rays [1]. The water and mud of the Dead Sea are extremely high in salinity and contain sulfides, microorganisms, algae, and other bioactive materials that may contribute to the therapeutic effect of Dead Sea water and mud [1]. A study on 1408 patients with AD found that 90% of patients had complete clearance of lesions after four to six weeks of therapy at the Dead Sea [97,117,118]. Avène thermal spring water (ATSW) has been known for its therapeutic effects since the middle of the 18th century [88]. ATSW is a natural water of deep origin with a constant physicochemical composition [123]. It has been classified in the category of waters with low mineral content. It is characterized by the presence of calcium, magnesium, and bicarbonate and significant amounts of silicates. ATSW originates by re-emergence from deep-in-the-ground rainwater. It infiltrated into the subsoil at least forty years ago (as measured by a tritium isotopic assay). The water migrates very slowly upwards from the depths towards the surface in a geological layer of dolomite (calcium magnesium carbonate) and dolomitic limestones [123]. Pharmacological and pharmacoclinical evidence from studies showed several effects that could explain the healing effect of ATSW: effect on histamine release, anti-inflammatory effects on standardized models, immuno-modulation of some cytokines involved in DA physiopathology (interferon (INF), interleukin 2 and 4 (IL-2, IL-4)), improvement of keratinocyte differentiation, and the effect on the skin microbiome by promoting the development of a diversified non-pathogenic flora [64,88,123]. In addition, an original microorganism, *Aquaphilus dolomiae*, never described in another medium, has very recently been identified in the ATSW [88]. *Aquaphilus dolomiae* is responsible for significant pharmacological activities on inflammation, pruritus, and enhancement of innate immunity. Many studies carried out at Avène hydrotherapy center have demonstrated efficacy in improving AD after a three-week course of treatment and a positive impact on patients’ quality of life [124,125,126]. Biometric analysis was performed to confirm that this improvement was due to the hydrotherapeutic treatment. Different enzymes (b-glucocerebrosidase, phospholipase A2, proteases, and cholesterol sulfatase) play an important role in the formation and maintenance of the epidermal barrier function in AD [127]. A positive correlation was demonstrated between the clinical stage of AD and the expression of these three enzymes [128]. Moreover, the three-week treatment had a positive impact on enzyme activities in atopic patients as they returned to activities like those observed in healthy subjects [129]. Recently, 210 AD patients were included in a comparative clinical study to determine if spraying ATSW contained in an over-the-counter product in addition to emollient applications in a skincare program could optimize the efficacy of the emollient [130]. Patients were divided into two groups: the first group applying a standardized emollient twice a day for 28 days (*n* = 107), and a second group using the same emollient twice a day for 28 days plus ATSW spray (*n* = 103). The severity of AD was reflected by SCORAD (Scoring atopic dermatitis): calculation was significantly improved in both the groups. The intensity of subjective signs of SCORAD (pruritus and sleep loss) was reduced by 63% in the group using emollient + ATSW and by 55% in the group using the emollient alone, with a significant difference between the two groups in favor of emollient + ATSW (*p* = 0.02). Thus, the association of ATSW spray with the application of an emollient improves the quality of life of patients with AD [126]. La Roche-Posay is a small town in the center of France that is famous for its dermatologic thermal treatment centers, the first of which is its namesake, La Roche-Posay Thermal Center [1]. In 1797, Napoleon Bonaparte had a thermal center built at the site of the thermal springs to treat his soldiers’ skin diseases [1]. La Roche-Posay TSW is a selenium-rich water (0.053 mg/L) that contains bicarbonate, calcium, silicate, magnesium, and strontium [107]. BT consists of an 18-day treatment (three weeks) with daily application of a high-pressure filiform shower (15 bars for three minutes), performed by a dermatologist. A study of 88 AD patients treated with La Roche-Posay BT had significant improvement of disease clinical manifestations, with an improvement of two severity scores, Eczema Area and Severity Index (EASI) and Dermatology Life Quality Index (DLQI), as well as improvement in itching and xerosis. Skin and quality-of-life benefits were maintained for an average of six months [131]. Finally, thermal BT at Comano spa (Trentino, Italy) has long been used in the treatment of various dermatologic conditions, with special regard for PSO and AD [104,132]. Comano water is an oligometallic thermal water, containing various microelements, among which calcium and magnesium are more represented; it has a temperature of 27 °C in the springs and a pH of 7.5–7.6 [104]. One controlled study investigates the efficacy and safety of BT performed at Comano spa (Trentino, Italy) compared to topical corticosteroids (TCS) for the treatment of pediatric AD [133]. BT with thermal spring water has been shown to be beneficial in children with mild to moderate AD with an effect similar to mid-potency topical corticosteroids [133].

### 4.3. Other Inflammatory Skin Diseases

Pruritus and prurigo: several approaches use water in complementary itch therapy, including balneology, heliotherapy, climate therapy, and mud use. Thermal sites, especially those around the Dead Sea and in other parts of the world that use mineral waters, mud, and soaks, as well as available sunlight, find perfect parallels in the clinical practice of dermatology with phototherapy, baths, and tar. Bathing and balneotherapy can relieve many types of itching, especially senile pruritus and seborrheic itching; chronic prurigo in adults, which is notoriously difficult to treat, can also respond to thermal therapy. The lesions present on the skin of these patients consist of scratches and can benefit from the antiseptic properties of some thermal waters [134]. 

Shani et al. reported that [117] two types of prurigo have been treated in the Dead Sea. Prurigo nodularis and prurigo simplex. Classically, there are two approaches for the treatment of this disease: the use of anti-inflammatory agents (steroids, salicylates, vasoactive drugs, etc.) and the use of agents of the central nervous system (sedatives, etc.); so balneotherapy can represent an alternative approach in patients with some contraindications for these systemic, complex therapies. In fact, balneotherapy has a calming effect and can reduce the doses or the duration of drug treatment. Balneotherapy at the Dead Sea area is also related to a reduction in itching for atopic patients during the first week of their stay at the site [117]. In conclusion, balneotherapy, as well as other complementary treatments, probably will increase its role in difficult-to-treat patients, and its indications continue to be better described year after year. 

Lichen ruber planus is a frequent, subacute or chronically progressive inflammatory papular skin disease that causes intense itching; it has some specific clinical and histological features. The treatment of lichen planus is based on the use of corticosteroids, antimalarial drugs, retinoids, and PUVA. Although balneotherapy and climatotherapy are not widely accepted for the treatment of lichen planus, all patients treated at the Dead Sea area showed significant improvement in their skin conditions and a reduction in itching during the four to six weeks of treatment, as reported by Shani et al. in 1997 [117]. In addition, in lichen ruber planus, balneotherapy can be alternated or combined with the drug therapy that often required a lot of time to achieve good clinical results. Also, the psychological support offered by the climate and the environment should not be underestimated in these patients, where treatment is often not simple and protracted for many years [134]. 

Ichthyoses are a group of diseases affecting keratinocyte differentiation; they can be differentiated on the basis of clinical, genetic, histopathological, ultrastructural, or biochemical features. In most of these conditions, abnormal keratinization involves the entire skin. The treatment of these diseases is exclusively symptomatic and consists of hydrating the skin, adding lipids to the damaged skin, and promoting its flaking [135]. Balneotherapy in water consisting of rather high quantities of sodium chloride is particularly useful for this purpose. Contraindications for balneotherapy in patients with ichthyosis include hypersensitivity to mineral baths, varicose veins, unhealed wounds, and acute and subacute dermatitis. Balneotherapy has the advantage that baths can be combined with sunlight, phototherapy, or other local or systemic therapies [117].

Balneotherapy and therapeutic thermal baths offer a healthy and relaxing atmosphere: as in other dermatological conditions, when the improvement is not totally achieved through the specific chemical and mineral components of the water, the benefits of relaxation and stress relief should not be underestimated in these patients [118]. There are also some studies about the efficacy of balneotherapy in association with phototherapy in lamellar ichthyoses and in ichthyosis linearis circumflexa [136,137].

Acne vulgaris is a common chronic inflammatory disease of the skin, that affects the pilosebaceous units and may result in inflammatory or non-inflammatory lesions. It has been researched extensively with regard to the disease itself as well as available and potential treatment options. The target for acne therapy is the four well-known pathogenic factors responsible for this disease: increased sebum production, irregular follicular desquamation, *Cutibacterium acnes* proliferation, and inflammation [138]. Immunomodulant and anti-inflammatory properties of thermal water are one of the determining factors involved in the therapeutic potential of thermal waters in acne. In fact, despite the conventional classification of acne lesions into inflammatory and non-inflammatory, evidence shows that inflammation is involved even in the earliest stages of acne pathogenesis [139]. Moreover, clay minerals can be used to treat acne, blackheads, and spots. A warm mixture of water and clay can facilitate pilosebaceous orifices opening and stimulate perspiration, inhibiting sebaceous secretions. Dead Sea black mud showed marked antimicrobial action when test microorganisms (*C. acnes*) were added to the mud where they lost their viability. Additionally, when Dead Sea mud was placed on *C. acnes* inoculated agar plates, a growth inhibition zone was observed [140,141]. Moreover, sulfuric thermal water, due to the sulfur keratolytic effect, resulting in peeling, can be therapeutic in acne patients contrasting follicular obstruction [33]. 

Due to the various pathological factors involved in acne development, the use of multimodal therapy which targets different processes simultaneously has been receiving considerable attention. Thermal water and balneotherapy can be considered as a valid option for combination therapy in acne patients, especially in the summer season when most topical and systemic acne therapies are contraindicated.

Seborrheic dermatitis is an inflammatory skin disorder characterized by red skin patches covered in oily and yellowish squamae and various degrees of itchiness, mainly found in seborrheic skin areas with many active sebum glands. Although the pathophysiological understanding of this condition is still incomplete, it is widely recognized the role of *Malassezia* yeasts in triggering an inflammatory and hyper-proliferative epidermal response. The treatment of seborrheic dermatitis aims to relieve inflammation, to suppress skin bacterial and fungal proliferation, and decrease sebum production. Therefore, the anti-inflammatory, antibacterial, and cheratolitic activity of thermal water has a high potential effectiveness for this condition [130]. Bathing in hypertonic salt solution promotes the removal of skin fat and scales, while bathing and local washing combined with sunlight can be recommended [34].

## 5. Discussion

Therapeutic effects of thermal waters are due to the combination of chemical, physical, immunological, and microbiological properties.

The dermatologic diseases that are frequently treated by balneotherapy with a high rate of success are PSO and AD.

Other skin diseases such as pruritus and prurigo, lichen ruber planus, acne vulgaris, and seborrheic dermatitis can benefit from the anti-inflammatory and antiproliferative activity of thermal water. 

The main importance of balneotherapy and thermal therapy, both as a monotherapy and as complementary therapy, lies in their potential efficacy after the failure of standard medical treatments.

More evidence to identify specific therapeutic algorithms for the use of thermal water in different skin diseases is needed. 

## Figures and Tables

**Table 1 jcm-09-03047-t001:** Chemico-physical properties: classification of thermal waters.

Classification of Thermal Water		
Classification According to Temperature	Classification According to Fixed Residue at 180 °C	Classification According to Chemical Composition
- cold (temperature below 20 °C) - hypothermal (20–30 °C) - homeothermal (30–40 °C) - hyperthermal (40–50 °C)	- Very low mineral content waters (fixed residue 600 mg/L) - Low mineral content waters (fixed residue 50–500 mg/L) - Medium mineral content waters (fixed residue 500–1500 mg/L) - Rich mineral content water (fixed residue 1 mg/L)	- Bicarbonate waters (>600 mg/L) - Carbonate waters (>300 mg/L) - Sulfated waters (>200 mg/L) - Sulfurous waters (>1 mg/L) - Arsenical-ferruginous waters (>1 mg/L) - Iodo-Bromo-Saline waters (>1 g/L) - Saline waters or Sodium-Chloride Rich waters (>200 mg/L) - Radioactive waters (>1 nC */L)

* Nanocurie.

**Table 2 jcm-09-03047-t002:** Potential effects on the skin of chemical elements present in thermal waters.

Thermal Water in Dermatology		
Type of Water	Most Abundant Chemical Elements	Beneficial Effects
Sulfated waters Sulfurous waters Bicarbonate waters	Magnesium, Sulfur	Skin regeneration, anti-inflammatory effects, and bactericidal activities
Iodo-Bromo-Saline waters Saline waters Carbonate waters	Calcium	Skin protection by improving natural defenses
	Manganese, Iodo, Bromo	Antioxidant effects
	Potassium	Skin hydration and enhancement of elastic tissues
Arsenical-ferruginous waters	Iron and Zinc	Replenishment of oxygen to the cells of the skin

**Table 3 jcm-09-03047-t003:** Prevalent bacterial phyla in thermal spring water based on temperature gradient.

Temperature	Thermal Spring Water (TSW)	Prevalent Phylum	**References**
>60 °C	Yellowstone National Park hot TSW Nakabusa hot TSW Bourlyashchy hottest TSW Chilas and Hunza hot TSW Tibetan hot TSW	Aquificae	Hall, et al., 2008; Everroad et al., 2012; Chernyh et al., 2015; Amin et al., 2017; Wang et al., 2013
<60 °C	Nakabusa hot TSW Tibetan hot TSW Yellowstone National Park hot TSW	Cyanobacteria (Thermosynechococcus/Synechococcus) Chloroflexi Proteobacteria	Everroad et al., 2012; Wang et al., 2013; Miller et al., 2009
20–35 °C	La Roche Posay TSW Gellért bath Lake Hevitz TW	Proteobacteria (alpha, beta, gamma) Actinobacteria Bacteroidetes Firmicutes	Zeichner et al., 2018; Szuróczki et al., 2016; Krett et al., 2016

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
