# Peer review of "The Role of Thermal Water in Chronic Skin Diseases Management: A Review of the Literature"

_jcm, 2020, doi:10.3390/jcm9093047_

Round 1

Reviewer 1 Report

This article on the the role of thermal water in chronic skin diseases management is a timely and important one, nicely prepared with a superb historical perspective. Balneotherapy has a calming effect on a variety of ailments and may reduce the doses or the duration of medication treatment. Balneotherapy at the Dead Sea or in spas such as a certain famous hotel in Budapest is popular for variety of conditions including psoriasis and atopic dermatitis. 

I concur that the main importance of balneotherapy and thermal therapy, both as monotherapy and as complementary therapy, lies in their potential efficacy after a failure of standard medical treatments. The dermatologic diseases that are frequently treated by balneotherapy with a high rate of success are psoriasis and atopic dermatitis. To me a large component of successful balneotherapy and thermal therapy is stress reduction. In fact, the authors note that balneotherapy and therapeutic thermal baths offer a health and relaxing atmosphere, observing that the "benefits of relaxation and stress relief should not be underestimated in these patients." 

Author Response

Thanks for your revision

Reviewer 2 Report

I think that the paper, in general, presents a nice revision of the literature. However some contributions are missing. The criteria is not clear as using the words and searching strategy described by the authors other published papers are found that were not included in this revision. Authors shall complete the revision or better describe the criteria for work selection.

Revision of the english is requiered and some mistakes and typing errors must be corrected:

  • line114 - theraeutical
  • Line 297 - The efficacy of TB for PSO is further underlined by other randomized controlled trials.22,23 
  • microrganisms´names shall be wirtten in italicum
  • P. acnes - is now called Cutibacterium acnes

In the section 3.3. Microbiological properties - the authors limit to discuss the microbial composition of the water. However, one of the important effect of thermal waters is the antimicrobial activity against epidermal pathogens related to skin diseases. I think that this perspetive shall be discussed also in this document. Relevant contribution can be found in the literature. check Environ Geochem Health. https://doi.org/10.1007/s10653-019-00473-6 

In the section 4.3 check the research paper Effect of São Pedro do Sul thermal water on skin irritation. M. O. Ferreira P. C. Costa M. F. Bahia
First published: 04 May 2010 https://doi.org/10.1111/j.1468-2494.2010.00527.x

Author Response

Point 1: I think that the paper, in general, presents a nice revision of the literature. However some contributions are missing. The criteria is not clear as using the words and searching strategy described by the authors other published papers are found that were not included in this revision. Authors shall complete the revision or better describe the criteria for work selection.

Response 1: According to the Reviewer’s suggestion, we explain which articles are excluded (lines 79-80).

Point 2: Revision of the english is requiered and some mistakes and typing errors must be corrected:

  • line114 - theraeutical
  • Line 297 - The efficacy of TB for PSO is further underlined by other randomized controlled trials.22,23 
  • microrganisms´names shall be wirtten in italicum
  • P. acnes - is now called Cutibacterium acnes

Response 2: We thank the Reviewer for the comment

  • Theraeutical: correct (line 116)
  • The efficacy of TB for PSO is further underlined by other randomized controlled trials.22,23: correct (line 307)
  • microrganism are written in italicum
  • P. acnes - is now called Cutibacterium acnes (section 4.3)

Point 3: In the section 3.3. Microbiological properties - the authors limit to discuss the microbial composition of the water. However, one of the important effect of thermal waters is the antimicrobial activity against epidermal pathogens related to skin diseases. I think that this perspetive shall be discussed also in this document. Relevant contribution can be found in the literature. check Environ Geochem Health. https://doi.org/10.1007/s10653-019-00473-6 

Response 3: We thank the Reviewer for the valuable comment. In the section 3.3 the antimicrobical effect of thermal waters against epidermal pathogens related to skin diseases was discussed considering relevant literary contributions(lines 238 to 245).

Point 4: In the section 4.3 check the research paper Effect of São Pedro do Sul thermal water on skin irritation. M. O. Ferreira P. C. Costa M. F. Bahia
First published: 04 May 2010 https://doi.org/10.1111/j.1468-2494.2010.00527.x

Response 4: We thank the Reviewer for the comment. We have added the discussion of this topic in the introduction (line 54).

Reviewer 3 Report

Comprehensive review and interesting historic background information on spring water bathing. I expect the paper is of interest to the readers.

Author Response

Thanks for your revision